# Comparison of new and emerging SARS-CoV-2 variant transmissibility through active contact testing. A comparative cross-sectional household seroprevalence study

**Katherine M. Gaskell** [1]☯*, **Natalie El Kheir**[1]☯, **Mariyam Mirfendesky**[2], **Tommy Rampling**[3], **Michael Marks**[4], **Catherine F. Houlihan**[5], **Norbert Lemonge**[6], **Hannah Bristowe**[6], **Suhail Aslam** [6], **Demetra Kyprianou**[2], **Eleni Nastouli**[7], **David Goldblatt** [8], **Katherine Fielding**[8], **David A. J. Moore**[1], **CONTACT team (field team)**¶

1 Clinical Research Department, Faculty of Infectious and Tropical Diseases, London School of Hygiene and Tropical Medicine, London, United Kingdom, 2 North Middlesex University Hospital NHS Trust, London, United Kingdom, 3 Hospital for Tropical Diseases, University College London Hospitals Foundation NHS Trust, London, United Kingdom, 4 Division of Infection and Immunity, University College London, London, United Kingdom, 5 Department of Clinical Virology, University College London Hospitals Foundation NHS Trust, London, United Kingdom, 6 Department of Population, Policy and Practice, University College London Institute of Child Health, London, United Kingdom, 7 Infection, Immunity & Inflammation Department, University College London; Great Ormond Street Institute of Child Health, London, United Kingdom, 8 Infectious Disease Epidemiology Group, London School of Hygiene and Tropical Medicine, London, United Kingdom

☯ These authors contributed equally to this work.
¶ The complete membership of this author group can be found in the Acknowledgments
* kate.gaskell@lshtm.ac.uk

**Data Availability Statement:** All relevant data are within the paper and its Supporting Information files.

## Abstract

Historically SARS-CoV-2 secondary attack rates (SAR) have been based on PCR positivity on screening symptomatic contacts; this misses transmission events and identifies only symptomatic contacts who are PCR positive at the time of sampling. We used serology to detect the relative transmissibility of Alpha Variant of Concern (VOC) to non-VOC SARS-CoV-2 to calculate household secondary attack rates. We identified index patients diagnosed with Alpha and non-VOC SARS-CoV-2 across two London Hospitals between November 2020 and January 2021 during a prolonged and well adhered national lockdown. We completed a household seroprevalence survey and found that 61.8% of non-VOC exposed household contacts were seropositive compared to 82.1% of Alpha exposed household contacts. The odds of infection doubled with exposure to an index diagnosed with Alpha. There was evidence of transmission events in almost all households. Our data strongly support that estimates of SAR should include serological data to improve accuracy and understanding.

## Introduction

In autumn 2020 the first SARS-CoV-2 variant of concern (VOC) B.1.1.7 was reported in Southeast England. National incidence was tracked by Public Health England through the

**Funding:** KMG Grant number: DONAT15914 (CONTACT) COVID-19 Response Fund Grants London School of Hygiene and Tropical Medicine https://www.lshtm.ac.uk/supportus/support-lshtms-covid-19-response-fund; Wellcome Trust with grant number 210830/Z/18/Z. The funders had no role in study design, data collection and analysis, decision to publish, or preparation of the manuscript.

**Competing interests:** The authors have declared that no competing interests exist.

failure of routine SARS-CoV-2 PCRs to detect the spike gene (spike gene target failure, SGTF) in community samples. Initial modelled estimates of the infectiousness of B.1.1.7. (subsequently named Alpha) ranged between 43–100% more transmissible than the previously circulating wild type SARS-CoV-2 virus [1, 2].

One empiric metric of transmissibility for infectious diseases is the secondary attack rate (SAR) in exposed contacts. To date, reporting of SAR in SARS-CoV-2 has mostly defined secondary infection as contemporaneous PCR positivity in a symptomatic contact [3]. This approach significantly underestimates the total number of people infected by only capturing those who are both symptomatic and still have detectable viral RNA at the time of sampling.

Historically, reported SARS-CoV-2 secondary attack rates based upon PCR-positivity were approximately 20% overall, irrespective of variant [4], though this was reported to rise to around 40% for Omicron [4, 5].

Serological testing captures previous infection, whether symptomatic or not, including potentially missed transmission events from which the contact has become PCR-negative by the time of sampling. Comparison of SARs between settings or viral variants should thus incorporate serological responses if the true transmissibility is to be understood. In a study from Singapore, PCR testing of symptomatic household contacts (HHC) found a SAR of 5.9% (95%CI 4.9–7.1), but Bayesian modelling of serological data estimated 62% of infections were missed [6].

In March 2021, four months into a national 'lockdown', we set out to compare the measured transmissibility of the Alpha variant of SARS-CoV-2 relative to other contemporaneously circulating variants (referred to as non-VOC) using serology and aimed to determine household SAR from index patients diagnosed with SARS-CoV-2 across two London Hospitals between November 2020 and January 2021.

## Methods

We performed a cross sectional seroprevalence study of HHC of individuals who had SARS-CoV-2 detected and whose virus was successfully whole genome sequenced (WGS) between 9th November 2020 and 31st January 2021. This period was selected as both Alpha and non-VOC SARS-CoV-2 were concurrently circulating within the community. Symptomatic index patients who were diagnosed across two North London hospitals (Hospital #1 and Hospital #2) were invited to participate alongside all their HHC. Potential participants were identified from those who submitted any sample where SARS-CoV-2 was detected by PCR to either laboratory including those submitted from the Emergency Department, out of hours GP services, occupational health services and hospitalised patients. We collected telephone verbal consent and undertook a questionnaire, subsequently visiting households once to collect serum with written consent for serological testing from all consenting household members. Verbal consent was witnessed by CONTACT team members and documented within the questionnaire. All household members of any age were eligible to participate. We initiated recruitment on 26th March 2021 for hospital #1 index cases and on 11th May 2021 for hospital #2. Recruitment was completed by 11th July 2021. The questionnaire included data on the index case, age, gender, ethnicity, and severity of COVID-19 disease, and for each contact on the duration of household exposure to the index patient whilst they were symptomatic, intensity of contact, additional SARS-CoV-2 exposures, vaccination status, and history of COVID-19 diagnosis or SARS-CoV-2 detection.

### Patient and public involvement

Patient and public engagement was sought at the local ethical approvals stage. The in-hospital patient advice and liaison service, the palliative care service and patient representatives were

involved in how best to approach patients, relatives, and relatives of the deceased. The research question, study design and outcome measures were not altered by this engagement, but operational research procedures were adapted to incorporate patient and relative priorities and preferences. Patients were not involved in the recruitment or conduct of the study. Participant serology results were sent to all who had a serum sample taken and the final results will be emailed to all participants on publication.

## Laboratory analysis

Samples were analysed for presence of IgG to SARS-CoV-2 spike protein (S) and nucleocapsid protein (NC) for both non-VOC and Alpha SARS-CoV-2 using a multiplex chemiluminescence immunoassay (MSD, Rockville, MD) evaluated by our laboratory as previously described [7]. All households were provided with their results.

## Statistical analysis

Assuming a conservative 15% seroprevalence in non-VOC households [8, 9] and an average cluster size of 2 (for a household size of 3; it is 2.4 in southeast England) we estimated that 292 Alpha and 292 non-VOC households (1168 participants) would provide 90% power to detect a 50% increase in seropositivity. For the primary outcome of interest, we used IgG reactivity to SARS-CoV-2 NC rather than IgG reactivity to S to confirm previous SARS-CoV-2 infection to avoid any effect of vaccination status. We subsequently undertook a secondary analysis with additional inclusion of unvaccinated IgG SARS-CoV-2 S reactive and IgG NC unreactive individuals to the primary outcome and assigned them as infected contacts (S3 Fig in S1 Appendix). We fitted a multivariable random effects logistic regression model adjusted for clustering at the household level having initially assessed the inclusion of continuous variables using fractional polynomials. We considered IMD rank, interval between index PCR diagnosis and HHC serum sampling and duration, and intensity of exposure to an infectious household index case as potential confounders and adjusted for these in our multivariable model (STATA code included in supplementary material).

## Definitions

HHC were defined as individuals living in the same household as the index case at the time of PCR-confirmed SARS-CoV-2. Duration of exposure was defined as the number of days of unquarantined exposure that an HHC had with the index case from the onset of symptoms up to hospital admission or symptom resolution. Whilst this unavoidably excludes pre-symptomatic infectiousness of index cases there is no evidence to suggest that this differs between VOC and non-VOC infections. Symptom duration captured the number of days the index was symptomatic at home. Socioeconomic status was captured through the use of the Index of Multiple Deprivation (IMD) which assigns nationwide ranking by postcode in the UK. Data were analysed by decile and rank. Long COVID was captured if participants had a clinical diagnosis of Long COVID.

## Ethics

The study was approved by the London School of Hygiene & Tropical Medicine Ethics Committee (LEO ref:25265), the NHS Health Research authority (IRAS ref:295376), and local hospital review committees. Verbal informed consent was obtained during the telephone survey and written consent provided prior to phlebotomy. Parents provided written consent for

children under 10 and assent was collected for children over 10 years. The study was sponsored by LSHTM.

## Results

During the study period a total of 1366 individuals tested positive for SARS-CoV-2 across the two sites and had viral genotyping data available. Of these 354 index participants agreed to take part and ultimately 238 index participants (50 non-VOC and 188 Alpha) and 454 household contacts completed the study questionnaires and had a serum sample available (Fig 1).

We collected blood samples from 102 HHC exposed to an index with non-VOC SARS-CoV-2 infection and 352 HHC exposed to an index with Alpha SARS-CoV-2 infection. The characteristics of the 386 participants who declined a serum sample but provided survey information did not differ significantly from those providing serum (S1 Table in S1 Appendix).

The characteristics of index cases are shown by index case Alpha/non-VOC assignment, in Table 1. 95% of index cases reported having been symptomatic, 76% reported respiratory symptoms and 67% were hospitalised, for a median of 9 days (IQR 4–21). There were no statistically significant differences between index cases with non-VOC and Alpha SARS-CoV-2 infection (Table 1).

Overall, the HHC study population was 56% female with a median age of 42 years (IQR 23–59). Median household size was three (IQR 2–4, range 2–9) with 46% of index cases reporting their ethnicity as white, 18% as black African or black Caribbean and 13% as south Asian. Local demographic data from the 2021 UK Census reported a population which is 52% female, with a median age of 36 years, a median household size of two, and 54% reporting their ethnicity as white, 21% as black African or black Caribbean, and 10% as Asian, indicating that the study participants were representative of the population from which they were drawn. Compared to contacts from non-VOC index households, HHC of Alpha index cases reported a greater average duration of exposure to their infectious index case, and to report a diagnosis of symptomatic COVID-19 around the time of their index case's diagnosis (Table 2).

### Spectrum of clinical features in seropositive contacts in Alpha and non-VOC affected households

A diagnosis of SARS-CoV-2 contemporaneous to the index case diagnosis was reported by 47% and 67% of contacts exposed to non-VOC and Alpha index cases within the household

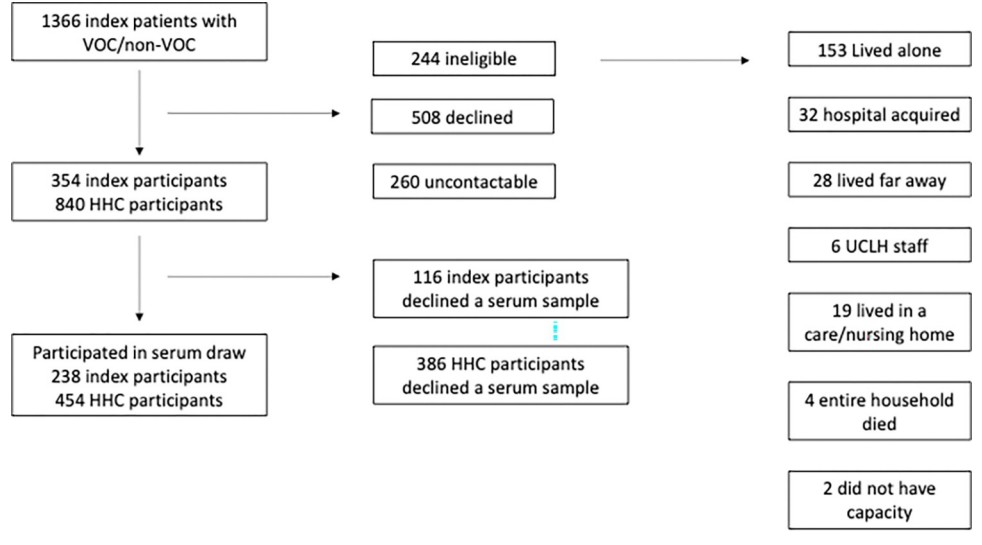

**Fig 1. Flow diagram for recruitment into the study.**

**Table 1. Baseline characteristics of 238 index cases stratified by index case Alpha or non-VOC SARS-CoV-2 status.**

| | | Non-VOC index case household | Alpha index case household |
|---|---|---|---|
| | Number of index—n | 50 | 188 |
| **Index characteristics** | | | |
| | index case female—n (%) | 26 (52) | 111 (59) |
| | Index case age–median (IQR) | 56 (45–72) | 57 (47–69) |
| ethnicity | white (%) | 27 (54) | 96 (51) |
| | Asian (%) | 2 (4) | 14 (8) |
| | Black (%) | 7 (14) | 34 (18) |
| | Middle Eastern (%) | 5 (10) | 10 (5) |
| | SE Asian (%) | 7 (14) | 13 (7) |
| | other | 2 (4) | 20 (10) |
| Hospital site | Hospital 1 | 20 (40) | 88 (47) |
| | Hospital 2 | 30 (60) | 100 (54) |
| | index case respiratory symptoms[1] –n (%) | 35 (70) | 146 (78) |
| | symptom duration in days–median(IQR) | 7 (3–10) | 7 (5–14) |
| | index case hospitalisation–n (%) | 29 (58) | 124 (66) |
| | index case ICU admission–n (%) | 5 (17) | 35 (29) |
| | index case mortality–n(%) | 1 (2) | 2 (1) |

[1] any of cough, dyspnoea, flu-like symptoms

respectively (48/102 vs. 235/352, chi squared p value = 0.02). Not all contacts who reported a diagnosis of COVID were symptomatic at the time, however either COVID symptoms or a SARS-CoV-2 diagnosis around the time of index case diagnosis were more frequently reported by contacts of an Alpha index case (OR 4.87, 95%CI 3.67–21.18); anti-nucleocapsid IgG titres were higher in those reporting symptoms (Fig 2).

## Seroprevalence in household contacts

61.8% (63/102) of non-VOC SARS-CoV-2 exposed HHC were seropositive compared to 82.1% (289/352) of Alpha SARS-CoV-2 exposed HHC. Household contacts exposed to an index with the Alpha variant had a 3.5-fold increased odds of being nucleocapsid seropositive (OR 3.5, 95%CI 1.7–7.4) when adjusted for household clustering.

The percentage of households in which no contacts had demonstrable SARS-CoV-2 anti-nucleocapsid antibodies, implying a complete absence of household transmission, was 3/50 (6%) and 1/188 (0.5%) in non-VOC and Alpha affected households respectively (Fisher's exact p value = 0.008).

Several covariates were identified as confounders requiring adjustment in logistic regression analyses, including IMD rank, interval between index PCR diagnosis and HHC serum sampling and duration and intensity of exposure to an infectious household index case (Table 3).

Non-VOC strains of SARS-CoV-2 presented in index cases earlier within the study period and earlier within the pandemic wave. Index cases with Alpha SARS-CoV-2 presented later within the pandemic wave and became the dominant strain causing COVID-19. The interval between index PCR diagnosis and HHC serological testing differed between the Alpha and non-VOC groups, (S1 Fig in S1 Appendix, Table 4). The observed decay in anti-nucleocapsid IgG titres over time (S2 Fig in S1 Appendix) highlights the importance of accounting for this differential delay; longer delay increases the risk of decay of anti-NC IgG below the threshold

**Table 2. Baseline characteristics of 454 household contacts stratified by index case Alpha or non-VOC SARS-CoV-2 status.**

| | | Non-VOC index case household | Alpha index case household |
|---|---|---|---|
| | Number of contacts—n | 102 | 352 |
| **Index Characteristics** | | | |
| | HHC exposed to a female index—n (%) | 58 (57) | 197 (56) |
| | Index age HHC exposed to—median (IQR) | 62 (45–76) | 56 (47–69) |
| HHC exposed to index ethnicity | white (%) | 47 (46) | 163 (46) |
| | Asian (%) | 6 (6) | 24 (7) |
| | Black (%) | 11 (11) | 72 (21) |
| | Middle Eastern (%) | 14 (14) | 19 (5) |
| | SE Asian (%) | 21 (21) | 39 (11) |
| | other | 3 (3) | 35 (10) |
| Hospital site | Hospital 1 | 49 (48) | 173 (49) |
| | Hospital 2 | 53 (52) | 179 (51) |
| | household size–median (IQR) | 3 (2–4) | 3 (2–4) |
| | HHC exposed to index case respiratory symptoms[1] –n (%) | 71 (70) | 276 (78) |
| | HHC exposure to symptomatic index in days–median (IQR) | 7 (4–12) | 7 (4–12) |
| | HHC exposure to a hospitalised index–n (%) | 62 (61) | 244 (69) |
| | HHC exposure to an index requiring ICU admission–n (%) | 11 (19) | 86 (35) |
| | HHC exposure to an index who died–n (%) | 8 (8) | 33 (9) |
| | Household IMD decile (IQR) | 3 (2–6) | 3 (2–6) |
| | Time since index PCR diagnosis—median (IQR) | 146 (125–181) | 119 (93–157) |
| **Contact characteristics** | | | |
| | contact female–n (%) | 57 (56) | 204 (58) |
| | contact age–median (IQR) | 44 (22–59) | 42 (24–60) |
| | days of exposure to index–median (IQR) | 4 (1–10) | 7 (3–14) |
| proximity to index[2] –n (%) | no close contact | 19(19) | 49(14) |
| | assisted in personal care | 24 (24) | 130 (37) |
| | shared bedroom | 16 (16) | 59 (17) |
| | shared bathroom | 43 (42) | 114 (32) |
| | other COVID exposure (not index)–n (%) | 26 (26) | 107 (30) |
| | COVID-19 diagnosis–n (%) | 48 (47) | 235 (67) |
| | Contact symptoms—n (%) | 40 (39) | 183 (52) |
| | Long COVID[3]—n (%) | 4 (4) | 8 (2) |
| vaccination status at time of serum sampling[4] | unvaccinated | 49 (48) | 169 (48) |
| | single vaccination | 24 (24) | 102 (29) |
| | double vaccination | 29 (28) | 81 (23) |

[1] any of cough, dyspnoea, flu-like symptoms

[2] this variable captures the intensity of contact—level '0' no close contact but still being a household member; level '1' sharing a bathroom but not sharing a bedroom; level '2' sharing a bedroom but not providing physical assistance; level '3' ascribed to those providing physical assistance to the unwell index case including washing, dressing, feeding, assisting with movement

[3] individuals with a clinical diagnosis of 'Long COVID'

[4] no more than 2 vaccinations available at this time

of detection and consequent misclassification of infected HHCs as seronegative, uninfected subjects.

Table 4 highlights the difference in median follow up time according to detection of IgG to S and NC, stratified by vaccination status (vaccination generates anti-S but not anti-NC IgG antibodies). The falling IgG NC titre over time did impact the odds of seropositivity in HHCs.

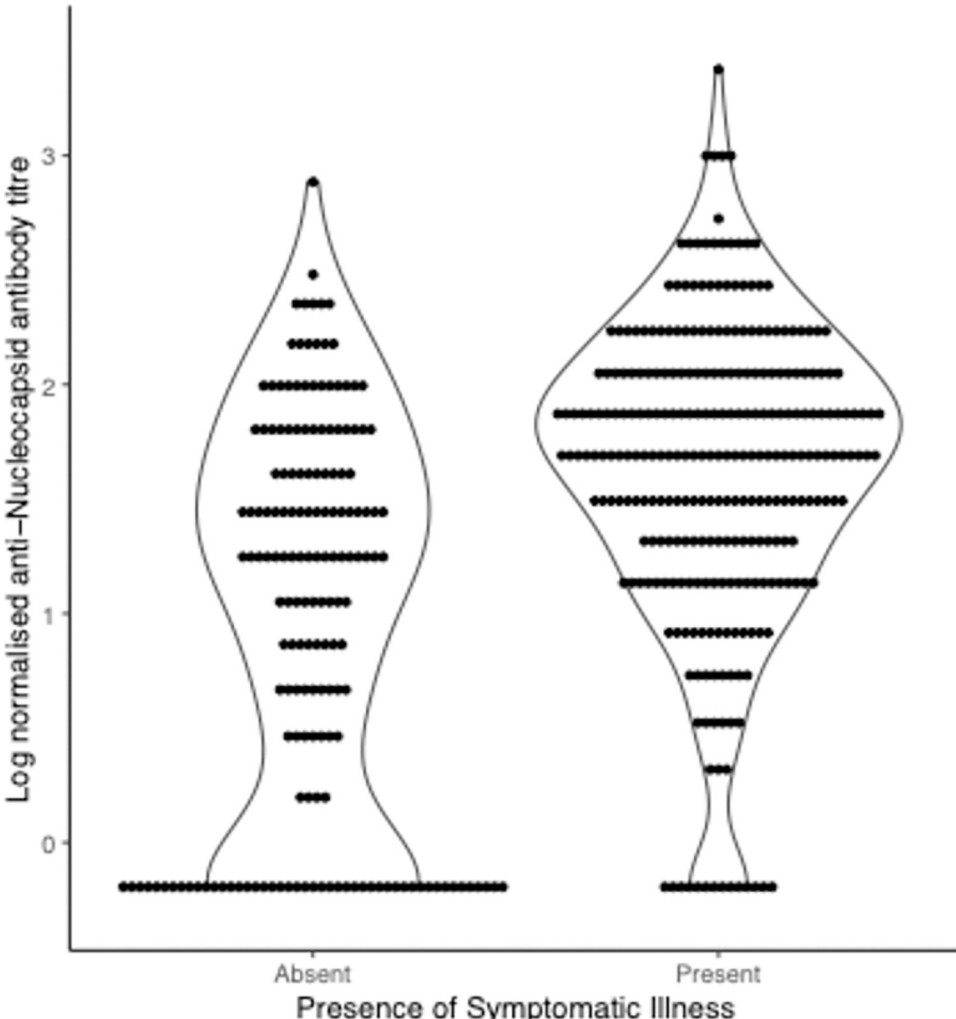

**Fig 2. Magnitude of anti-nucleocapsid SARS-CoV-2 antibody response in 454 household contacts with and without reported COVID symptoms.**

This contrasts with previous published work where nucleocapsid titre declines more rapidly over the first 6 months from infection than Spike protein titre but does not impact positivity overall.

**Table 3. Unadjusted and adjusted odds ratios for the effect of index case Alpha SARS-CoV-2 on household contact infection (defined as anti-nucleocapsid IgG sero-positivity) compared to non-VOC.**

| | % NC positive (n/N) | Unadjusted OR (95% CI) | P-value | Adjusted OR*(95% CI) | P-value |
|---|---|---|---|---|---|
| **Non-VOC** | 62% (63/102) | | | | |
| **Alpha** | 82% (289/352) | 3.5 (1.7–7.4) | 0.001 | 2.5 (1.2–5.4) | 0.02 |

All OR are adjusted for household level clustering

* adjusted for sex, age (years; linear), time between index PCR date and date of HHC serology (days; linear), duration of contact between infectious index and HHC (days; linear), IMD (rank; linear), intensity of contact (4 levels assuming a linear trend).

NC = nucleocapsid, n = total positive, N = total population

**Table 4. Relationship between vaccination status, antibody positivity and the interval between household exposure (PCR positive test in index case) and serological testing of HHC.**

| Vaccination status | Serological response | Number of contacts | Interval between household exposure and serological testing | |
|---|---|---|---|---|
| | | Total n = 454 | median | IQR |
| Unvaccinated Total 221 | Spike positive | 166 | 117 | 91–147 |
| | NC positive | | | |
| | Spike positive | 32 | 157 | 121–161 |
| | NC negative | | | |
| | Spike negative | 3 | 198 | 143–198 |
| | NC positive | | | |
| | Spike negative | 20 | 124 | 108–167 |
| | NC negative | | | |
| Vaccinated Total 233 | Spike positive | 183 | 133 | 108–161 |
| | NC positive | | | |
| | Spike positive | 50 | 137 | 126–163 |
| | NC negative | | | |
| | Spike negative | 0 | | |
| | NC positive | | | |
| | Spike negative | 0 | | |
| | NC negative | | | |

To improve the classification of infection and explore the effect of Alpha exposure further we performed a secondary analysis using a combined outcome definition of infection including all NC IgG positive contacts and unvaccinated participants seropositive for anti-S IgG (S3 Fig in S1 Appendix) we found a similar effect to in our main analysis. We performed an unadjusted and an adjusted logistic regression model with this outcome, adjusting for the same confounding exposures, which yielded similar though slightly lower point estimates (aOR 2.1, 95%CI 0.8–5.5) (Table 5).

## Discussion

The most striking finding of our study is the very high secondary infection rate amongst household contacts of SARS-CoV-2 irrespective of variant; 62% in non-VOC exposed and 82% in Alpha exposed contacts. The odds of infection in a household contact, already high with the original SARS-CoV-2 virus, doubled with the arrival of Alpha. Very few households were identified in which no transmission had taken place (6% and 0.5% of non-VOC and Alpha households respectively). A recently published human challenge model of pre-Alpha

**Table 5. Unadjusted and adjusted odds ratios for the effect of index case Alpha variant SARS-CoV-2 on household contact infection (defined as either anti-nucleocapsid IgG seropositivity or anti-spike IgG seropositivity in an unvaccinated individual) compared to non-VOC SARS-CoV-2.**

| | % infection (n/N) | Unadjusted OR (95% CI) | P-value | Adjusted OR*(95% CI) | P-value |
|---|---|---|---|---|---|
| **Non-VOC** | 75% (77/102) | | | | |
| **Alpha** | 87% (307/352) | 3.1 (1.2–8.3) | 0.02 | 2.1 (0.8–5.5) | 0.14 |

All OR are adjusted for household level clustering

* adjusted for sex, age (years; linear), time between index PCR date and date of HHC serology (days; linear), duration of contact between infectious index and HHC (days; linear), IMD (rank; linear), intensity of contact (4 levels assuming a linear trend)

n = total positive, N = total population

SARS-CoV-2 identified viral shedding onset within 2 days of infectious exposure continuing up to 12 days post-inoculation [10]. Our data demonstrate a dose-response of exposure to transmission, supporting the suggestion that being able to isolate from household members mitigates against transmission of SARS-COV-2 to some extent, regardless of the strain or variant.

These data, capturing evidence of infection previously overlooked by highly time-sensitive estimates of SAR dependent upon PCR testing of contacts, demonstrate much higher rates of infection than have been previously reported [11]. In addition to the effect of Alpha, household contacts were much more likely to be infected with greater duration and higher intensity of exposure in a dose-response fashion. Whilst Omicron is believed to have outcompeted and cause more infections than previous VOCs, due in part to a high rate of re-infections in the partially immune [11], Danish data reported a household secondary attack rate using RT-PCR of only 31% and 21% in Omicron and Delta SARS-CoV-2 infections respectively [12]. A meta-analysis of household SARs for Alpha and Delta VOC were estimated at 38% and 31% respectively [4], significantly lower than our study findings. Most previous data included secondary cases identified as RT-PCR positive infections only [12]. A previous study assessing a composite secondary attack rate using both serology and PCR results in Los Angeles found a similarly high SAR of 77% to our study [13]. Our data strongly support that estimates of SAR should include serological data to improve accuracy and understanding.

Our original analysis focussed upon anti-nucleocapsid IgG as the marker of infection as earlier data had suggested this should remain detectable within the timeframe that our study was planned for. In the event, recruitment was delayed longer than expected and some anti-NC IgG decay was seen earlier than previously reported [11]. To capture those who we believed to have been infected but in whom anti-NC IgG was undetectable (or no longer detectable at the threshold of the test) we took advantage of the unvaccinated contacts with only anti-S IgG to improve our definition of infection. The lower adjusted odds ratio of 2.1 is thus more likely to be closer to the true value.

Our study has several limitations. Firstly, an assumption of all our analyses is that the most likely explanation for a household contacts' seropositivity is their household exposure. It is plausible that exposure may have occurred during the first wave of infections in England which started in March 2020, or indeed outside the household at any time including to a common source at the same time as the index. Any such effect would tend to exaggerate the odds in Alpha contacts, because Alpha infections came later, allowing for greater exposure-risk time. However, low population prevalence prior to November 2020 and societal lockdown conditions in England will have minimised this effect, although adherence to lockdown was dwindling during the emergence of Alpha. We have no local data for community prevalence during the period studied.

Secondly, the index cases in this study were tested and diagnosed in hospital laboratories so may have been more severely unwell than the general population. Whilst this might bias overall estimates of the SAR, we did not find any evidence of difference in severity between the two exposure groups and so this is unlikely to have impacted our finding of an increased SAR amongst contacts of Alpha infection. The increased severity may have resulted in higher rates of viral shedding and more prolonged viral shedding, however those with more severe illness in this population were admitted to hospital, interrupting domiciliary transmission. We cannot present data on the viral loads or cycle thresholds (CT) for PCR positive index patients as a variety of PCR platforms were used across the laboratories involved, some of which reported relative light units. There are reported differences between variants in viral loads, duration of shedding, and in duration of infectivity [14–16]; our data cannot be directly extrapolated to more recent VOCs.

Finally, we were unable to meet our planned sample size within the available time frame to complete the study prior to relaxation of lockdown regulations, in part due to delays in regulatory and ethics application approvals. Community uptake and participation with the study was lower than anticipated both at initial approach and for the follow up serum sample. Despite this we were still able to demonstrate the increased SAR for Alpha compared to non-VOC infections and that using serology resulted in a markedly higher estimate of the SAR for both VOC and non-VOC infections.

Secondary attack rates amongst household contacts are a direct measure of transmissibility, particularly if community transmission is not high. There is however a risk that saturation—almost all contacts becoming infected—can obscure differences between infecting strains. Careful participant selection minimises the risk of confounding by time, place or person allowing the rates of secondary infection and secondary disease to be directly estimated and compared between different virus variants. Calculating transmissibility for novel and previous VOC(s) is an essential component of the public health response. These data support our argument to include serology in calculating secondary attack rates to assess transmissibility of new SARS-CoV-2 VOCs more accurately. Household contact screening to assess SAR is a useful model to replicate in future variant-driven pandemic waves.

## Conclusions

Secondary attack rates (SAR) in SARS-CoV-2 were previously calculated using PCR positive samples only, though it is more accurate to use a household transmission model and screen contacts using serology, as done in this study. SAR should include serological data to improve accuracy and understanding. Almost all households in this study had transmission events. SAR were 61.8% in non-VOC SARS-CoV-2 exposed household contacts compared to 82.1% in Alpha SARS-CoV-2 exposed household contacts.

## Supporting information

**S1 Appendix.**
(DOCX)

**S1 File.**
(PDF)

**S1 Data.**
(XLSX)

## Acknowledgments

The authors acknowledge all the participants and their patients involved in this research and the entire CONTACT field team for their hard work throughout this project who were all affiliated with the London School of Hygiene and Tropical Medicine. The CONTACT field team include Suhail Aslam, Jaimie Wilson-Goldsmith, Nuwoe L S Tehmeh, Henrietta Bristowe, Daniel Osigbe, Edward Monk, Meskerem A Kebede, Norbert Lemonge, Sarah Wookey.

## Author Contributions

**Conceptualization:** Natalie El Kheir, Mariyam Mirfendesky, Tommy Rampling, Michael Marks, Catherine F. Houlihan, Eleni Nastouli, David Goldblatt, Katherine Fielding, David A. J. Moore.

**Data curation:** Katherine M. Gaskell, Natalie El Kheir, Hannah Bristowe, Suhail Aslam.

**Formal analysis:** Katherine M. Gaskell, David Goldblatt, Katherine Fielding.

**Funding acquisition:** Katherine M. Gaskell.

**Investigation:** Katherine M. Gaskell, Natalie El Kheir, Mariyam Mirfendesky, Tommy Rampling, Catherine F. Houlihan, Norbert Lemonge, Hannah Bristowe, Suhail Aslam, Demetra Kyprianou, David Goldblatt, Katherine Fielding, David A. J. Moore.

**Methodology:** Katherine M. Gaskell, Natalie El Kheir, Mariyam Mirfendesky, Tommy Rampling, Michael Marks, Catherine F. Houlihan, Eleni Nastouli, Katherine Fielding, David A. J. Moore.

**Project administration:** Katherine M. Gaskell, Natalie El Kheir, Norbert Lemonge, Hannah Bristowe, Suhail Aslam, Demetra Kyprianou.

**Resources:** Mariyam Mirfendesky, Tommy Rampling, Catherine F. Houlihan, Eleni Nastouli, David Goldblatt, David A. J. Moore.

**Supervision:** Katherine M. Gaskell, Michael Marks, David Goldblatt, Katherine Fielding, David A. J. Moore.

**Validation:** Katherine M. Gaskell, Natalie El Kheir, Michael Marks, David Goldblatt, Katherine Fielding, David A. J. Moore.

**Writing – original draft:** Katherine M. Gaskell.

**Writing – review & editing:** Natalie El Kheir, Mariyam Mirfendesky, Tommy Rampling, Michael Marks, Catherine F. Houlihan, Norbert Lemonge, Hannah Bristowe, Suhail Aslam, Demetra Kyprianou, Eleni Nastouli, David Goldblatt, Katherine Fielding, David A. J. Moore.

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
