## [Decision Letter · Decision Letter 0]

23 Nov 2022

PONE-D-22-28360Comparison Of New and emerging SARS-CoV-2 variant Transmissibility through Active Contact Testing. A comparative cross-sectional household seroprevalence study.PLOS ONE

Dear Dr Gaskell,

Thank you for submitting your manuscript to PLOS ONE. After careful consideration, we feel that it has merit but does not fully meet PLOS ONE’s publication criteria as it currently stands. Therefore, we invite you to submit a revised version of the manuscript that addresses the points raised during the review process.

We look forward to receiving your revised manuscript.

Kind regards,

Dr Gayathri Delanerolle

Academic Editor

PLOS ONE

Journal Requirements:

2.Please provide additional details regarding participant consent. In the ethics statement in the Methods and online submission information, please ensure that you have specified what type you obtained (for instance, written or verbal, and if verbal, how it was documented and witnessed). If your study included minors, state whether you obtained consent from parents or guardians. If the need for consent was waived by the ethics committee, please include this information.

4. Thank you for stating the following in the Affiliation Section of your manuscript: 

"This research was funded by the LSHTM COVID-19 response fund Grant number DONAT15914 and performed independently to any intervention by this funder. This research was funded in part by the Wellcome Trust, [210830/Z/18/Z]. For the purpose of Open Access, the author has applied a CC BY public copyright licence to any Author Accepted Manuscript (AAM) version arising from this submission."

"KMG

Grant number: DONAT15914 (CONTACT)

COVID-19 Response Fund Grants London School of Hygiene and Tropical Medicine

https://www.lshtm.ac.uk/supportus/support-lshtms-covid-19-response-fund

NO - The funders had no role in study design, data collection and analysis, decision to publish, or preparation of the manuscript"

6. One of the noted authors is a group or consortium CONTACT team (field team). In addition to naming the author group, please list the individual authors and affiliations within this group in the acknowledgments section of your manuscript. Please also indicate clearly a lead author for this group along with a contact email address.’ 

Reviewers' comments:

Reviewer's Responses to Questions

**Comments to the Author**

1. Is the manuscript technically sound, and do the data support the conclusions?

Reviewer #1: Yes

2. Has the statistical analysis been performed appropriately and rigorously? 

Reviewer #1: I Don't Know

3. Have the authors made all data underlying the findings in their manuscript fully available?

Reviewer #1: No

4. Is the manuscript presented in an intelligible fashion and written in standard English?

Reviewer #1: Yes

5. Review Comments to the Author

Reviewer #1: See attached PDF. I am writing nonsense in this text box because it will not let me submit the review unless I type a minimum of 200 characters. This is the longest 200 characters I have ever written.

6. PLOS authors have the option to publish the peer review history of their article (what does this mean?). If published, this will include your full peer review and any attached files.

Reviewer #1: No

---

## [Author Response · Author response to Decision Letter 0]

15 Mar 2023

Dear Editor, 

Thank you for sharing the comments and observations from the review of our manuscript entitled:

Comparison Of New and emerging SARS-CoV-2 variant Transmissibility through Active Contact Testing. A comparative cross-sectional household seroprevalence study. 

In this letter we address and respond to each point, reproduced in full here, in turn:

1. This is a very nice study that demonstrates convincingly the higher transmissibility of the alpha variant of SARS-CoV-2 as compared to the non-VOC strain. The data supporting this comes from a serological survey which is extremely useful as it eliminates bias that comes from only counting symptomatic individuals as is the usual practice. My main concern with this article is that the methodology is barely explained at all nor is any software code available. Though the general technique is standard, a more thorough explanation is required because there can be devils in those details: in this case the obvious one is how censorship is handled. It would also be useful to report SITP alongside SAR.

We are grateful to the reviewer for these kind comments. The queries are addressed below.

2. There is neither an explanation of the statistical methodology (e.g. in math) nor any code made available that I could find from which to understand precisely how the analysis was carried out. I would expect both to be provided. It is essential to provide this information because without it there is no way to understand what statements like "OR are adjusted for household level clustering" mean and likewise how censoring (see next point) was handled. It isn't enough to say that the code is only useful with the data and the data is only available upon request.

By way of further explanation, we have added the following underlined wording to the statistical analysis section of the methods. ‘We subsequently undertook a secondary analysis with additional inclusion of unvaccinated IgG SARS-CoV-2 S reactive and IgG NC unreactive individuals to the primary outcome and assigned them as infected contacts (supplementary figure 3). We fitted a multivariable random effects logistic regression model adjusted for clustering at the household level having initially assessed the inclusion of continuous variables using fractional polynomials. We considered IMD rank, interval between index PCR diagnosis and HHC serum sampling and duration, and intensity of exposure to an infectious household index case as potential confounders and adjusted for these in our multivariable model (STATA code included in supplementary material)’

Additionally, at the request of the reviewer we have included the STATA code shown below as supplementary information in the appendix. We did not feel that it fitted logically within the main body of the text though we would be prepared to move it back in if the Editors felt this was necessary.

Supplementary information. 

STATA code for primary analysis

mfp, df(agenew diffpcrsero dayshh imd_rank close_contact:4): xtlogit nc_pos vocnew sexnew agenew diffpcrsero dayshh imd_rank close_contact, re or nolog

mfp: xtlogit infected vocnew sexnew agenew diffpcrsero dayshh imd_rank close_contact, re or nolog 

3. Of the households for which an index case took part, it seems that not all contacts also took part and of those who did, not all had serology available. In other words, there is clearly censoring in the data. Indeed 454 contacts fully participated (line 169) and 386 declined serology but filled in a survey (line 174). It is unclear from the methodology how this censoring was handled in the analysis. We should be able to get upper (assume all of those who declined became infected) and lower (assume none of them became infected) bounds on the estimates of SAR from this.

We recognise the reviewer’s concern about contacts for whom serological testing was not available and the potential implications for the generalisability of the results. For precisely this reason, we explicitly included supplementary table 1 in the manuscript. This table compares the demographics and exposures of contacts who did and did not provide a serum sample. As we stated in lines 199-201 ‘The characteristics of the 386 participants who declined a serum sample but provided survey information did not differ significantly from those providing serum (supplementary table 1)’. Data were not ‘censored’ but rather inclusion depended upon availability of test data; generalisability of the results rests upon the demonstration (in supplementary table 1) that the included participants did not differ from those not included. The calculation of upper and lower bounds in this situation would not add meaningful information but would create confusion around the analysis so we have avoided this.

4. Though SAR is a standard measure in the field, SITP (susceptible-infectious transmission probability) is more useful especially when comparing across studies. Ideally this could be computed and reported for each household size. This is only a suggestion but I think it would make this study stronger.

The suggestion from the reviewer is interesting but we have not calculated the susceptible infectious transmission probability (SITP) as we did not include a transmission model within this data set and so do not have an estimate of the transmission rate with which to perform this calculation.

5. Definition of duration of exposure is somewhat problematic because it doesn't admit the possibility of presymptomatic transmission. However, I think it is likely that trying to fix this could signicantly complicate the analysis without substantially affecting the overall result. Suggest to simply mention this.

We acknowledge that this warrants mention and have included the following (underlined) wording into lines 161-163 ‘Duration of exposure was defined as the number of days of unquarantined exposure that a HHC had with the index case from the onset of symptoms up to hospital admission or symptom resolution. Whilst this unavoidably excludes pre-symptomatic infectiousness of index cases there is no evidence to suggest that this differs between VOC and non-VOC infections.’

6. Representativeness. Study population 56% female, median age 42years, etc. One would expect to report these figures alongside the corresponding ones for the population in the study geography as a whole to help judge the possibility of bias. Such bias, if present, could be attributable to sampling or to differential circulation of the virus in different subpopulations. It is unlikely to be possible to tell from the data collected but it is still important to know. These figures for the background demographics should be readily available so reporting them shouldn't be a significant burden.

 We thank the reviewer for this request and we are pleased to have included corresponding local data from the national census of 2021 indicating that the study population is representative of the population from which they are drawn. We have included the following text in lines 218-221 ‘Local demographic data from the 2021 UK Census reported a population which is 52% female, with a median age of 36 years, a median household size of two, and 54% reporting their ethnicity as white, 21% as black African or black Caribbean, and 10% as Asian, indicating that the study participants were representative of the population from which they were drawn.’

We trust that our responses have fully addressed any remaining queries.

Yours sincerely,

Dr Katherine M Gaskell

---

## [Editor Report · Decision Letter 1]

29 Mar 2023

Comparison Of New and emerging SARS-CoV-2 variant Transmissibility through Active Contact Testing. A comparative cross-sectional household seroprevalence study.

PONE-D-22-28360R1

Dear Dr Gaskell,

We’re pleased to inform you that your manuscript has been judged scientifically suitable for publication and will be formally accepted for publication once it meets all outstanding technical requirements.

Kind regards,

Dr Gayathri Delanerolle

Academic Editor

PLOS ONE

---

## [Editor Report · Acceptance letter]

11 Apr 2023

PONE-D-22-28360R1 

Comparison Of New and emerging SARS-CoV-2 variant Transmissibility through Active Contact Testing. A comparative cross-sectional household seroprevalence study. 

Dear Dr. Gaskell:

I'm pleased to inform you that your manuscript has been deemed suitable for publication in PLOS ONE. Congratulations! Your manuscript is now with our production department. 

Kind regards, 

on behalf of

Dr. Gayathri Delanerolle 

Academic Editor

PLOS ONE